# Cytokine Secretion Dynamics of Isolated PBMC after Cladribine Exposure in RRMS Patients

**DOI:** 10.3390/ijms231810262

**Published:** 2022-09-06

**Authors:** Rodica Balasa, Smaranda Maier, Adina Hutanu, Septimiu Voidazan, Sebastian Andone, Mirela Oiaga, Doina Manu

**Affiliations:** 1Ist Neurology Clinic, Emergency Clinical County Hospital, 540136 Targu Mures, Romania; 2Department of Neurology, ‘George Emil Palade’ University of Medicine, Pharmacy, Science, and Technology of Targu Mures, 540136 Targu Mures, Romania; 3Doctoral School, ‘George Emil Palade’ University of Medicine, Pharmacy, Science, and Technology of Targu Mures, 540142 Targu Mures, Romania; 4Department of Laboratory Medicine, ‘George Emil Palade’ University of Medicine, Pharmacy, Science, and Technology of Targu Mures, 540142 Targu Mures, Romania; 5Laboratory Medicine, Emergency Clinical County Hospital Targu Mures, 540136 Targu Mures, Romania; 6Department of Epidemiology, ‘George Emil Palade’ University of Medicine, Pharmacy, Science, and Technology of Targu Mures, 540136 Targu Mures, Romania; 7Anaesthesiology and Intensive Care Clinic, Emergency Clinical County Hospital Targu Mures, 540136 Targu Mures, Romania; 8Center for Advanced Medical and Pharmaceutical Research, ‘George Emil Palade’ University of Medicine, Pharmacy, Science, and Technology of Targu Mures, 540136 Targu Mures, Romania

**Keywords:** multiple sclerosis, cladribine, immunomodulation, PBMC secretory profile

## Abstract

Cladribine (CLD) treats multiple sclerosis (MS) by selectively and transiently depleting B and T cells with a secondary long-term reconstruction of the immune system. This study provides evidence of CLD’s immunomodulatory role in peripheral blood mononuclear cells (PBMCs) harvested from 40 patients with untreated relapsing-remitting MS (RRMS) exposed to CLD. We quantified cytokine secretion from PBMCs isolated by density gradient centrifugation with Ficoll–Paque using xMAP technology on a FlexMap 3D analyzer with a highly sensitive multiplex immunoassay kit. The PBMC secretory profile was evaluated with and without CLD exposure. PBMCs isolated from patients with RRMS for ≤12 months had significantly higher IL-4 but significantly lower IFN-γ and TNF-α secretion after CLD exposure. PBMCs isolated from patients with RRMS for >12 months had altered inflammatory ratios toward an anti-inflammatory profile and increased IL-4 but decreased TNF-α secretion after CLD exposure. CLD induced nonsignificant changes in IL-17 secretion in both RRMS groups. Our findings reaffirm CLD’s immunomodulatory effect that induces an anti-inflammatory phenotype.

## 1. Introduction

Multiple sclerosis (MS) is a demyelinating and neurodegenerative disease that mainly affects young adults. Over 2.5 million individuals are affected by this immune-mediated disease of the central nervous system (CNS) [1]. While MS remains incurable, numerous disease-modifying treatments (DMT) can slow its progression and accumulation of disabilities, improving longevity and quality of life.

MS is a heterogeneous disease from pathogenic, immunological, clinical, and DMT treatment response perspectives. Currently, the European Medical Agency (EMA) has approved 14 DMTs for MS treatment [2]. However, personalizing MS treatment has become a health priority problem due to distinct outcomes and different therapeutic responses among patients. In addition, after years of MS progression, the immunological profile of a given treated patient is dynamic, and reevaluation of the type of DMT used might be required. The priority in MS treatment is to identify the optimal treatment benefit for each patient [3,4]. Minimization of side effects is also a constant concern of MS specialists. All DMTs are, to some degree, immunotherapies, and the identification of immune biomarkers to predict treatment response represents an essential step in silencing MS activity. However, currently proposed predictors of responders (Res) vs. nonresponders (nonRes) for each immunotherapy used for MS patients remain contentious. Nevertheless, the effectiveness of treating MS patients with DMTs has been shown, and it is accepted that the earlier treatment is begun during the clinical MS course, the greater its effectiveness.

Given the autoimmune nature of MS, the role of immune cell subsets from T and B lymphocyte groups has been extensively studied at MS onset and during its progression [5]. As a cell-mediated disease, assessing the nature and dynamics of specific immune cell subpopulations might attenuate the response rates to different DMTs, since their mechanism of action preferentially target specific T cell subsets (e.g., natalizumab and S1P mediators) or decreases different B cell subpopulations (e.g., ocrelizumab and rituximab). Immune reconstruction therapy (IRT) uses drugs that transiently decrease lymphocytes in different subpopulations (i.e., partial immunosuppression) [6]. These treatments are followed by recovery and reconstruction of different beneficial immune structures. Cladribine (CLD) is an IRT drug that selectively targets 70–90% of B lymphocytes and up to 45% of T lymphocyte subtypes [7]. Numerous studies have explored the nature and the dynamic of reduction followed by reconstruction of lymphocytes after CLD treatment. The clinical benefit of CLD in clinical trials, such as Clarity, goes beyond its mild and short-lived T cell depletion and severe B cell depletion [8]. CLD immune reconstruction modifies the surviving immune cell types and causes a major shift in their secretory profile. Therefore, the mechanism of action and benefits of CLD treatment is still being studied [9].

However, approximately 50% of relapsing-remitting MS (RRMS) patients do not respond to CLD treatment [10,11]. NonRes patients are identified retrospectively after a minimum of one year of therapy, leading to the accumulation of neurological disabilities and financial wastage. CLD reportedly has a peripheral mechanism of action that includes an immunomodulatory effect with an anti-inflammatory shift in the T cell cytokine environment and lymphopenia in the CD4+, CD8+, and CD19+ (up to 90% of the CD19+ population) cell subpopulations [9,12]. However, due to its molecular size, CLD can pass through the blood-brain-barrier (BBB), decreasing oligoclonal band production and potentially targeting CD19+ B cells in lymphoid follicle-like structures in the meninges, influencing the chronic neurodegenerative process that evolves into progressive MS forms [6]. The beneficial effects observed in CLD-treated RRMS patients persisted for many months after the cessation of CLD treatment. Therefore, we hypothesize that in addition to its direct cytotoxic effects, CLD’s mode of action may involve immunomodulatory mechanisms. Similar to other immunosuppressive drugs, such effects might consist of alterations in cytokine patterns in surviving cells [8]. Because some lymphocytes secrete multiple types of cytokines, it is pertinent and timely to assess the variation in secretory cytokines profiles of cultured and stimulated peripheral blood mononuclear cells (PBMCs) harvested from RRMS patients with CLD exposure to enable improved risk/benefit evaluations for CLD treatment in MS patients in future.

The aim of this study was to evaluate the change in the cytokine secretion profile of PBMCs harvested from naïve RRMS patients in the presence/absence of CLD.

## 2. Results

Demographic and clinical data of the 40 naïve RRMS patients included in this study are presented in Table 1. The mean age of RRMS patients was 36.08 ± 9.59, similar to the mean age of HCs, 35.82 ± 8.71. The sex distribution was also similar between the two groups, with 27 (67.5%) RRMS patients female and 13 (32.5%) male compared to 13 (65%) and 7 (35%) for HCs, respectively. RRMs patients had relatively stable disease, with an annual relapse rate in the year prior to this study of 1.38 ± 1.00, and had been in a remitting period for at least 30 days. RRMS patients were grouped based on disease duration at PBMC collection, with 19 having had RRMS ≤ 12 months and 21 >12 months. The concentration of cytokines measured from the culture media and the inflammatory ratio for HCs and RRMS patients are presented in Table 2.

### 2.1. Secretory Profile of PBMCs Harvested from All RRMS Patients Compared to HCs

To assess the effect of CLD exposure on the cytokines secretory profile of PBMCs in the culture medium, we compared the levels of five cytokines and the inflammatory ratio between HCs and RRMS patients at the beginning of the study and on days 7 and 14 with or without CLD exposure. Based on Dunn’s multiple comparison test, we found:IL-4 (Table 3)
-IL-4 secretion was significantly increased in CLD− PBMCs from HCs on day 14 compared to day 7 (*p* = 0.0275), while the increase was not significant in CLD+ PBMCs (*p* = 0.3308).-IL-4 secretion was significantly increased in CLD− (*p* < 0.0001) and CLD+ (*p* < 0.0001) PBMCs from RRMS patients on day 14 compared to day 7.IL-17A
-IL-17A secretion was nonsignificantly decreased in CLD+ but nonsignificantly increased in CLD- PBMCs from HCs on day 14 compared to day 7.-IL-17A secretion was nonsignificantly increased in CLD− PBMCs from RRMS patients on day 14 compared to day 7 but significantly increased in CLD+ PBMCs on day 14 (9.20) compared to day 7 (5.99).TNF-α (Table 4)
-TNF-α secretion was significantly increased in CLD+ (*p* = 0.0440) and CLD− (*p* = 0.0074) PBMCs from RRMS patients on day 14 compared to day 7, but not in HC-harvested cells (*p* > 0.3233).IL-10 and IFN-γ secretion and the inflammatory ratio did not differ significantly.

### 2.2. Secretory Profile of PBMCs Harvested from RRMS Patients with Disease Duration ≤ 12 Months

We found significantly higher IL-4 secretion on day 14 in CLD+ PBMCs compared to CLD− PBMCs (*p* = 0.0015). In contrast, IL-10 (Figure 1) and IL-17A (Figure 2) secretion were significantly decreased on days 7 and 14 in both CLD+ and CLD- PBMCs compared to ex vivo. When we compared IL-17A secretion on days 7 and 14, we found it to be increased in both CLD+ and CLD- PBMCs, with a slightly and nonsignificantly greater increase in CLD+ PBMCs.

IFN-γ and TNF-α secretion decreased significantly on day 14 only in CLD+ PBMCs (*p* < 0.0001) but did not change significantly in CLD− PBMCs. In addition, TNF-α secretion by CLD+ PBMCs was nonsignificantly lower (8.13 pg/mL) than by CLD− PBMCs (11.64 pg/mL) on day 14. No statistically significant differences in the inflammatory ratio were observed.

### 2.3. Secretory Profile of PMBCs Harvested from RRMS Patients with Disease Duration > 12 Months

The secretory profile of PBMCs from RRMS patients with disease duration > 12 months showed significantly increased IL-4 secretion on day 14 compared to day 7 without (*p* = 0.0089) and with (*p* = 0.0023) CLD exposure. However, CLD exposure did not significantly alter IL-10 and IL-17A secretion. Dunn’s multiple comparison test indicated a statistically significant decrease in IL-10 and IL-17A secretion in CLD- and CLD+ PBMCs (Figure 3 and Figure 4). By comparing IL-17A secretion between days 7 and 14, we found it was increased in CLD- PBMCs (5393 pg/mL and 6174 pg/mL, respectively) but nonsignificantly decreased in CLD+ PBMCs (14.62 pg/mL and 5.260 pg/mL, respectively).

TNF-α secretion on day 7 was not influenced by CLD exposure, although a statistically significant decrease compared to ex vivo was observed, which remained statistically significant on day 14 only in CLD+ PBMCs (Table 5). CLD exposure also changed the inflammatory ratio in RRMS patients with disease duration > 12 months, which decreased significantly on day 14 compared to day 7 in CLD+ PBMCs (*p* = 0.0415).

## 3. Discussion

Major advances in MS treatment consist not only of developing new drug molecules but also of characterizing and measuring the pathological immune response during treatment and for MS prognosis [13]. From disease mechanisms to clinical applications, MS treatment can provide insight into MS pathogenesis, such as the newly described mode of action of a DMT identifying a novel MS pathogenic mechanism.

RRMS is mainly characterized by T-cell-mediated demyelination involving subtypes that substantially produce IL-10, IL-17, and IFN-γ [1]. Therefore, these cytokines can be used as biomarkers of differential MS activity during different disease stages and to monitor the immune system response to DMTs. Heterogeneity in the immunologic pathway influences responses to DMTs, as seen in the cytokine profiles of RRMS patients in previously published studies [14,15]. The pathogenic initiators of MS are periphery-activated CD4 T cells secreting cytokines such as IFN-γ, TNF-α, and IL-17 [13]. Other cytokines, such as IL-4 and IL-10, help to monitor the activity of T-cell subpopulations involved in MS pathophysiology. IL-10 is an immunomodulatory cytokine with predominantly suppressive actions produced by many cell types, including T regulatory cells [16,17,18].

CLD treatment of MS patients frequently provides a sustained reduction in clinical and MRI inflammatory disease activity without any rebound even after immune reconstruction [19]. The reduction in lymphocyte count is transient, and there is a minimal effect on innate immune function. After short-term CLD administration, there is a progressive reconstitution of the lymphocyte populations [20]. Despite a very intense mechanism of action, not all MS patients have their MS silenced with CLD treatment, or they become resistant (nonRes) to therapy [17]. Therefore, it is essential to make progress in initially selecting Res patients for certain DMTs, to select the best population and to avoid the risk of side effects of CLD administered to patients who will not benefit.

CLD has been found to have multiple mechanisms of action that complement its main effect of gradually depleting lymphocytes, contrasting with the rapid reductions seen with alemtuzumab, a mAb with a cytolytic mode of action [21,22,23,24]. These effects that persist long after treatment identify CLD as an immunomodulatory DMT that partially impairs immunity during treatment-free periods and acts in an anti-inflammatory manner. This effect suggests that partially known qualitative changes develop in the adaptative immune cells after exposure to CLD [12]. It is important to note that CLD penetrates the BBB, possibly potentiating microglial apoptosis and, by decreasing monocyte chemotaxis, have an anti-migratory Natalizumab-like effect by diminishing the penetration of circulating leucocytes into the CNS. In addition, CLD-induced immune reconstitution provides a long-lasting decrease in the intrathecal humoral response/oligoclonal bands, enhancing its therapeutic effect on RRMS progression. Moreover, CLD depletes other immune cells such as monocytes, dendritic, and natural killer cells that produce proinflammatory cytokines. Furthermore, CLD stimulates the differentiation of naïve T cells in the direction of T regulatory cells and other tolerogenic phenotypes that secrete anti-inflammatory cytokines IL-4, IL-5, and IL-10 [25,26,27,28].

The optimization of MS management might require both the initial isolation of PBMCs from MS patients and cytokine profiling with exposure to certain DMTs in the context of different treatments having different cell targets and altered cytokine secretory profiles.

Treatment with CLD tablets at 3.5 mg/kg selectively reduced B and T lymphocytes in the pooled data from CLARITY, CLARITY Extension studies, and the PREMIERE registry [11,29,30]. Our interest was drawn to T lymphocyte behavior after CLD exposure because they play an essential role in RRMS onset and progression. Approximately 45% of lymphocytes survive CLD exposure, and T lymphocyte recovery begins soon after CLD treatment ends. The pressing issue with CLD tablets is their effect on tissue-infiltrating T lymphocytes, mainly in the CNS [7]. Comi et al., evaluated long-term lymphocyte count changes in pooled data from the 2-year CLARITY study followed by 2-year CLARITY Extension studies and the PREMIERE registry, finding that the median CD4+ T cell counts recovered ~43 weeks after CLD treatment in year 2, but median CD8+ cell counts remained below their threshold value [12].

Some studies have found no correlation between RRMS activity in CLD-treated patients and the T cell subset depletion level [31]. Therefore, we selected naïve RRMS patients in the remitting period for this study.

Numerous studies have shown that the immunomodulatory effect of CLD treatment appears secondary to selective immunosuppression. Preclinical, clinical, in vitro, and in vivo studies found both reduction of proinflammatory cytokines and chemokines (IFN-γ and TNF-α) and elevation of anti-inflammatory cytokines (IL-4, IL-5, and IL-10) secreted both by leukocytes and by dendritic cells exposed to CLD [7,16].

For practical reasons, evaluating the presumed effect of a certain DMT in selected RRMS patients, such as changes in isolated lymphocyte immune behavior in the presence of the drug, is more accurate regarding the possible individual modulation of autoimmune mechanisms leading to MS progression than the isolation and characterization of each target cell. Immune cells might be extremely versatile in their secreted cytokine pattern. MS exacerbates in patients under very different developmental patterns as a manifestation of very different pathologic mechanisms. It is known that CLD produces lymphopenia due to B and T cell depletion. Therefore, we looked closer at the secreting characteristics of the remaining cells. Since there is a 45–50% decrease in CD4+T cells with CLD treatment, its efficacy must rely on other mechanisms of action, and immunomodulation might be one of them. The important role of CD4+T cells in MS pathogenesis has been known for a long time, and as a consequence, this cell population has been the most studied. However, most studies have used small numbers of RRMS patients and different methods for determining the cell secretion profile causing conflicting results [32].

We propose a more accessible and time-saving approach that represents a clear step toward personalized treatment in MS patients by determining the changes induced by a certain DMT on the cytokine secreting profile of PBMCs isolated from RRMS patients. Since immune cells are pluripotent in their secretory profile, it is more important to evaluate the possible Res status to a certain DMT than to the cell type itself. For example, T helper (Th) 17 cells were considered a subset of IL-17-secreting Th cells with a proinflammatory role mainly at the onset of MS [15,33]. However, further studies showed that Th17 cells could be stimulated to produce Th1 and Th2 cytokines. While pathogenic murine Th17 cells express higher levels of IFN-γ, non-pathogenic Th17 cells express IL-10 and IL-17. It is important to note that some cytokines might also be secreted by other cell types. For example, IL-17 can be produced by astrocytes and natural killer cells and IL-10 by B cells [17,34,35,36].

Importantly, the upregulated T cell group in MS patients is represented by Th17.1 double positive cells, expressing both IL-17 and IFN-γ. This cell group not only produces a powerful proinflammatory cytokine repertoire in the periphery but also has a direct deleterious effect on the BBB cells [18,37,38]. Our data did not show any statistical difference in the cytokine ratio of the two groups, suggesting that there are no immunologically distinct subgroups of MS patients (onset ≤ 12 vs. >12 months). The difference in disease onset of only one year likely has no influence on the studied cytokines secretion by PBMCs. In addition, we showed in a previous study that IL-17 serum concentrations were higher at MS onset and that IFN-beta treatment had potential differential effects depending on MS duration [15]. This group had a mean MS duration of 21 months before PBMC collection, and even when we divided the RRMS patients based on disease duration, we did not observe any significant effect of CLD exposure on IL-17 levels, indicating that CLD does not influence IL-17-secreting cells irrespective of the stage of MS progression. Another aspect is that our RRMS patients were in the remitting phase, and the levels of some cytokines, such as IL-10 and IFN-γ, were increased in this phase, while others, such as IL-4 and IL-17, were decreased in previous studies.

Instead of determining the level of cytokines secreted by PBMCs, we evaluated the secretory profile of isolated PBMCs, because we intended to project their behavior in the culture medium. The data were obtained in vitro, with an important success rate encouraging us to trust our cell isolation method. This method might be used in the treatment personalization of naïve MS cases since ILs have important roles in MS pathogenesis. Fissolo et al., also isolated PBMCs from RRMS patients and exposed them to CLD, investigating the impact of in vitro CLD exposure on the activation of PBMC subsets, finding that CLD’s immunomodulatory effect is via decreased immune cell proliferation and activation together with increased apoptosis of lymphocyte subsets. Like our results, their findings require in vivo confirmation in RRMS patients taking CLD [27]. Such in vivo confirmation was performed by Moser et al., who performed a longitudinal study on the depletion and restitution kinetics together with the PBMC cytokine profiles of 18 RRMS patients treated with CLD [39]. They found good depletion with CLD treatment not only of B lymphocytes but also of Th17 and Th17.1 cells. Recently, it was shown that Th17 cells display developmental plasticity, creating Th17.1 cells that secrete both IFN-γ and IL-17, which is very pathogenic in MS. These aggressive cells infiltrate the CNS early, and experimental autoimmune encephalomyelitis (EAE) models have shown some involvement in microglial activation with an important role in MS onset and CNS compartmentalization. The involvement of IFN-γ+ Th17 cells in MS pathology was described together with their preferential recruitment into the CNS during inflammatory events. The isolation method was very effective, but adding CLD to the culture did not decrease IL-17 secretion in any group, irrespective of CLD exposure duration. This finding is consistent with Dobreanu et al., (2021), who concluded that CLD targets the Th17 population [14,39,40,41,42], which was indirectly confirmed in this study by the nonsignificant change in IL-17 secretion in both groups after CLD exposure. Therefore, an initial increase in IL-17 secretion by PBMCs in RRMS patients could indicate CLD treatment avoidance.

The differences between our findings and those of Moser et al., might be explained by different MS duration populations (2 vs. 8 years, respectively), environment (in vitro vs. in vivo), and design (cross-sectional vs. longitudinal). In the study by Moser et al., the Th17 decrease was most pronounced in year two, suggesting that the efficacy of CLD may increase by the second cycle. However, we must note that they used the same methods for cell isolation and cultivation [39].

IL-4 is an important cytokine with a regulatory effect on immune cells and a protective role in CNS inflammation, including T and B cell stimulation, that is secreted mainly by Th2 lymphocytes. In this study, IL-4 secretion increased after CLD exposure in both HCs and RRMS patients, but the greatest increase was in RRMS patients with short disease durations, consistent with a previous study that found CLD significantly increased IL-4 concentrations in cultured PBMCs from healthy donors. Therefore, our findings indicate that CLD stimulates IL-4 production as an immunoregulatory role irrespective of disease duration [8,43].

PBMCs harvested from RRMS patients and exposed to CLD changed their secretory profile by decreasing TNF-α production. This cytokine is mainly produced by macrophages during acute inflammation but also by T CD4+ cells [44]. Our isolation method excluded macrophages from the cell culture. Consequently, any change in TNF-α secretion was due to the surviving T CD4+ cells. TNF-α involvement has been studied in diverse pathological hallmarks of MS, such as immune dysregulation, neuroinflammation, demyelination, and synaptopathy [45]. Mathiesen et al., evaluated CLD’s effect on monocyte differentiation into macrophages M1 and M2 in vitro, finding that the therapeutically relevant in-vitro concentrations reduced the TNF-α secretion after activation [46]. These findings provide a new perspective on CLD’s potential mode of action in which it passes through the BBB into the brain leading to oversecretion of TNF-α by glial cells, astrocytes, and microglia in pathological conditions such as MS.

In the early-tested MS group, CLD significantly decreased the secretion of proinflammatory cytokines such as IFN-γ in cells in an exposure-dependent manner. IFN-γ is produced by cells of the innate immune response, such as natural killer cells, and by CD4 Th1 and CD8 cytotoxic T lymphocytes in the adaptive immune response. Until recently, IFN-γ was considered a strict proinflammatory cytokine and the hallmark of Th1 cell activation. Therefore, it is interesting that recent studies have suggested it has a dual role in CNS cells depending on its concentration. Low doses induced protection in both microglia and oligodendrocytes, but high doses exacerbated MS effects in glial cells. In addition to this dose-dependent effect, other studies have found that the dual effect of IFN-γ also varies according to the stage of disease progression, having a protective effect in the chronic stages of MS [47,48,49,50,51,52,53].

Importantly, we showed that the inflammatory ratio shifted toward an anti-inflammatory slope with prolonged CDL exposure in RRMS patients with disease durations > 12 months. Other studies also found an inflammatory ratio shaped only by IL-4/IFN-γ. Korsen et al., isolated PMBCs from healthy donors, finding significant changes in IL-4 secretion but not TNF-α and IFN-γ secretion with CLD exposure, which might be of therapeutic benefit for identifying MS patients for CLD treatment [8].

The correlation of diverse serological factors before and after CLD treatment with in vivo studies might have some limitations. The absence of correlations with PBMC IL secretion activity might reflect the CNS compartmentalization of autoimmunity during MS outside the peripheral blood. However, immune cells entering the CNS might trigger disease activity. In addition, compartmentalization of the immune response inside the lymphoid tissues might also explain the absence of correlations [22]. Therefore, other lymphocyte secreted proteins beyond those considered here that might influence the immune response of RRMS patients should be considered [7].

Major limitations of this study include that the secretion of surviving cells in the presence of CLD in vitro will differ from in vivo due to the much higher CLD concentrations used in vitro than in patients and that the controlled in vitro conditions will differ from the more capricious in vivo environment. The exact mechanisms by which CLD exerts its beneficial effect in most RRMS patients remains unknown. However, qualitative and quantitative adjustments in lymphocyte subsets will be crucial in determining CLD’s effect.

Previous studies have found that different T cell subsets show different plasticity and that the expression of signature subsets of cytokine might change in vivo [54,55]. Consequently, finding a more reliable predictive biomarker for treatment response is important regardless of cost. 

## 4. Materials and Methods

### 4.1. Patients Selection

We performed a prospective noninterventional pilot study of 40 consecutive patients diagnosed with RRMS according to the Mc Donald 2017 criteria [56] during their remitting phase, naïve to any DMT, at the Regional Center for Multiple Sclerosis in Targu-Mures, Romania. In addition, we recruited 20 age- and sex-matched healthy controls (HCs) for comparison. This study was approved by the Ethics Committee of the Targu Mures County Emergency Clinical Hospital (decision number 7100/2018), and all experiments were performed according to the principles of the Helsinki Declaration. The recruitment of RRMS patients and HC took place between 1 March 2018, and 31 December 2018. The inclusion criteria for RRMS patients were: (1) RRMS diagnosis according to McDonald’s 2017 criteria; (2) naïve to any DMT; (3) Aged over 18; (4) No relapses or treatment with corticosteroids 30 days before collection of serum samples. Exclusion criteria were: (1) Refusal to participate; (2) Clinical or paraclinical signs of systemic infection; (3) Neoplastic disorders. All patients and HCs signed the informed consent form.

Demographic data (age and sex) were collected from RRMS patients and HCs. Disease-related information was also collected from RRMS patients: duration, number of relapses in the last year, and degree of disability assessed by the Expanded Disability Status Scale (EDSS).

### 4.2. Cell Isolation and Culture

Peripheral blood from naïve RRMS patients and HCs was drawn in heparinized tubes. We acquired an automated complete blood count (CBC) for each subject, and their high-sensitivity C-reactive protein (hsCRP) plasma level was assessed. Subjects with white blood cells/L values > 10 × 109 in CBC reports or hsCRP values > 3 mg/L were excluded. PBMCs were isolated by density gradient centrifugation with Ficoll-Paque. Following isolation, PBMCs were cryopreserved with 10% dimethyl sulfoxide (DMSO) at -140 °C. PBMCs were then thawed and cultured in Roswell Park Memorial Institute (RPMI)-1640 medium containing L-glutamine supplemented with 1% penicillin/streptomycin and 10% fetal bovine serum at a density of 1–2 × 106 cells/mL.

In the ex vivo analysis of cytokine secretion, PBMCs were activated for 72 h with soluble NA/LE CD3 monoclonal antibody (mAb) (BD, clone HIT3a, cat. no. 555336) at a final concentration of 1 µg/mL and CD28 mAb >(BD, clone 28.2, cat. no. 555725) at a final concentration of 5 µg/mL. Culture media were harvested and cryopreserved at −80 °C until required.

In the in vitro analysis, PBMCs were thawed and activated with CD3/CD28 mAbs for 72 h, then T cell receptor (TCR)-activated cells were incubated for 48 h with with 10^−7^ M (100 nM) CLD (Sigma Aldrich, cat. no. C4438) or without CLD. After CLD exposure, cells were resuspended in culture media supplemented with 10 ng/mL recombinant human interleukin 2 (rhIL-2) and cultured for 5 or 10 days. Next, rhIL-2 was removed 20 h prior to cell restimulation with CD3/CD28 mAbs on days 7 and 14. Then, the culture media was harvested and cryopreserved for further multiplex cytokine analysis. The protocol for CLD exposure was previously described by Korsen et al. [8]. CLD concentration was selected for an optimal cell response in term of viability and immunomodulatory effect.

A viability assay was performed using the FACSAria III flow cytometer and Becton Dickinson Horizon Fixable Viability Stain 780 (FVS 780, BD cat. no. 565388) before and after CLD exposure and at day 7 or 14. The cytotoxic effect of CLD was evaluated by absolute CBC using Sysmex XS 800i Hematology Analyzer. The changes in absolute lymphocyte number reflected cell proliferation or depletion in response to the cytotoxic action of CLD. The results concerning the cytotoxic effect of CLD on T cells were previously published by our team (Dobreanu et al. [14]. Here are our findings: the initial T cell proliferation after 48 h of CLD exposure was lower for treated cells from both HC and RRMS patients, compared to untreated cells. However, cells from RRMS patients proliferated better compared to HC. T cells of RRMS are more resistant to CLD compared to T cells of HC. In both HC and RRMS, only treated T cells proliferated continuously until day 14, suggesting that survival T cells, resistant to CLD display a consistent proliferative behavior.

The secreted cytokine profile was measured using xMAP technology on a FlexMap3D Luminex analyzer with a cytokine panel built with ProcartaPlex Multiplex Kits from Invitrogen (Human High-Sensitivity Panel 9-Plex; cat.no. EPXS090-12199-901). Data were acquired and analyzed with the xPONENT software, (ver 4.2) (Luminex Corporation Austin Texas, A DiaSorin Company, Saluggia, Italy). Cell culture and activation were performed according to the protocols described by Korsen et al. [8].

### 4.3. Cytokine Analysis

The supernatant was collected for cytokine quantification in the culture media, aliquoted, and stored at −80 °C until required. We used the ProcartaPlex Human High Sensitivity Panel from Invitrogen to quantify cytokines of interest from TCR-activated cells. This multiplex panel quantifies interferon-gamma (IFN-γ), tumor necrosis factor-alpha (TNF-α), interleukins (IL)-4, IL-10, and IL-17A with high sensitivity. The performance (sensitivity) characteristics of the kit included an upper and lower limit of quantification (ULOQ/LLOQ) according to the certificate of analysis provided by the manufacturer: 1475/1.44 pg/mL for IFN-γ, 4080/1.00 pg/mL for TNF-α, 4330/1.06 pg/mL for IL-4, 765/0.19 pg/mL for IL-10, and 770/0.19 pg/mL for IL-17A.

This protocol used magnetic microspheres coated with analyte-specific antibodies to enable the simultaneous quantification of multiple target proteins in small samples. The microspheres are internally dyed with specific fluorophores of various intensities, corresponding to different bead set regions. Briefly, the protocol consisted of incubating 50 µL of cell culture supernatant or standards (4-fold serial diluted) with 50 µL of bead mixture. After overnight incubation at 4 °C, the antigen-antibody complex will have formed on the surface of the corresponding beads. After a washing step, the antigen-antibody complexes on the bead surface were tagged by adding a biotinylated detection antibody followed by a streptavidin-phycoerythrin (SA-PE) conjugate.

After a final incubation and washing steps, the beads are resuspended and analyzed using the xMAP technology on a FlexMap 3D analyzer, a high-throughput platform for multiple analytes detection. The bead analysis is performed using two lasers, the red laser for bead classification according to their spectral signature and the green laser to evaluate the fluorescent reporter bound on the immune complexes captured on the bead’s surface. Data acquisition and interpretation were performed using the xPONENT software for Luminex instruments. By interpolating the median fluorescent intensity (MIF) of samples on the five parameter logistic calibration curves obtained within the same run, the software estimates the concentrations of each cytokine of interest. Based on the obtained cytokine values, the inflammatory ratio was calculated: (IL-17 + TNF-α + IFN-γ)/(IL-4 + IL-10). Study protocol is presented in Figure 5.

### 4.4. Statistical Analysis

Statistical analyses were performed with GraphPad v.3.6 (Dotmatics; San Diego, CA, USA). We assessed the normality of continuous variables with the Shapiro–Wilk test. The Student’s *t*-test was used for intergroup comparisons of continuous variables, which are reported as mean ± standard deviation (SD). The Mann–Whitney U test was used for two groups and Kruskal–Wallis tests was used for intergroup comparisons of nonparametric variables, which are reported as median and range (minimum–maximum), which can be used with >2 groups. Dunn’s multiple comparison test was used to assess groups between which there were statistically significant differences. All results with *p* < 0.05 were considered statistically significant.

## 5. Conclusions

Despite these limitations, this study is among the few that have used an exploratory method that enables the evaluation of CLD’s immunomodulatory effects on surviving PBMCs. Our findings are consistent with previous studies that could not explain the long-term effects of CLD only by direct cytotoxicity. These data suggest that cells surviving the well-known cytotoxic effect of CLD change their secretion toward an anti-inflammatory phenotype. The induction in CLD-treated RRMS patients of an anti-inflammatory cytokine pattern makes CLD a pluripotent drug with cytotoxic (mainly on B cells) and long-term immunomodulatory (mainly on surviving T cells) mechanism of action. CLD targets pathogenic T cells directly (cytotoxic effect) or indirectly (immunomodulatory effect), both mechanisms being well-known MS therapeutic strategies. This study addressed whether CLD exerts immunomodulatory effects on surviving immune cells along with its known cytotoxicity. The answer was yes, at least in vitro, since CLD induces the secretion of IL-4 and decreases the secretion of TNF-α and IFN-γ.

## Figures and Tables

**Figure 1 ijms-23-10262-f001:**
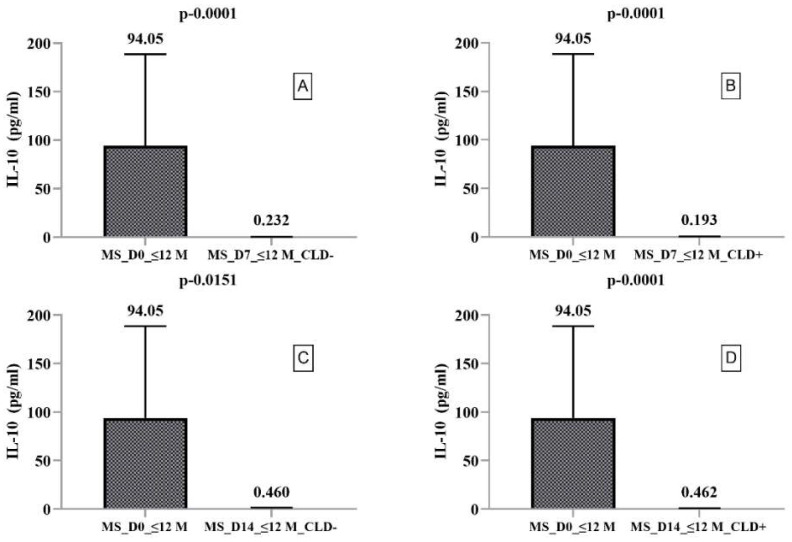
Boxplot type diagrams through which we followed the medians and interquartile range. Median values and statistical significance are mentioned in the graph. Comparison of CLD exposure effects on IL-10 secretion by PBMCs harvested from RRMS patients with a disease duration of ≤12 months on day 0 with days 7 and 14. (**A**) Comparison between the level of IL-10 secreted by PBMC ex vivo and on day 7 of the study in the absence of exposure to CLD; (**B**) Comparison between the level of IL-10 secreted by PBMC ex vivo and 7 days after exposure to CLD; (**C**) Comparison between the level of IL-10 secreted by PBMC ex vivo and on day 14 of the study in the absence of exposure to CLD; (**D**) Comparison between the level of IL-10 secreted by PBMC ex vivo and 14 days after exposure to CLD.

**Figure 2 ijms-23-10262-f002:**
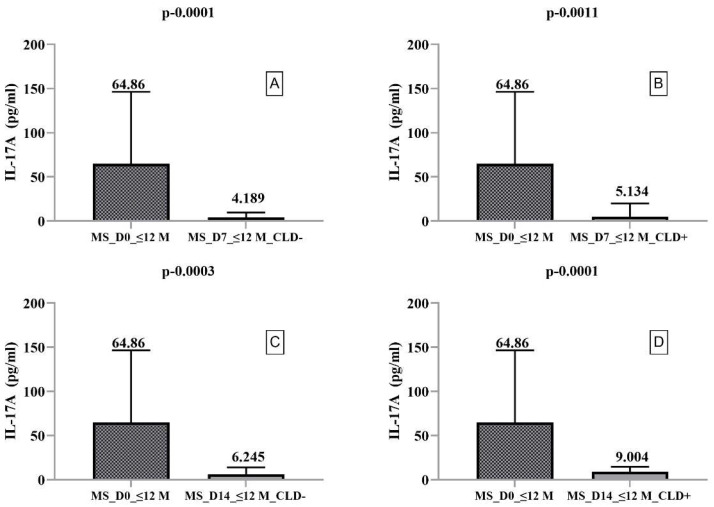
Boxplot type diagrams through which we followed the medians and interquartile range. Median values and statistical significance are mentioned in the graph. Comparison of CLD exposure effects on IL-17A secretion by PBMCs harvested from RRMS patients with a disease duration of <12 months on day 0 with days 7 and 14. (**A**) Comparison between the level of IL-17A secreted by PBMC ex vivo and on day 7 of the study in the absence of exposure to CLD; (**B**) Comparison between the level of IL-17B secreted by PBMC ex vivo and 7 days after exposure to CLD; (**C**) Comparison between the level of IL-17A secreted by PBMC ex vivo and on day 14 of the study in the absence of exposure to CLD; (**D**) Comparison between the level of IL-17A secreted by PBMC ex vivo and 14 days after exposure to CLD.

**Figure 3 ijms-23-10262-f003:**
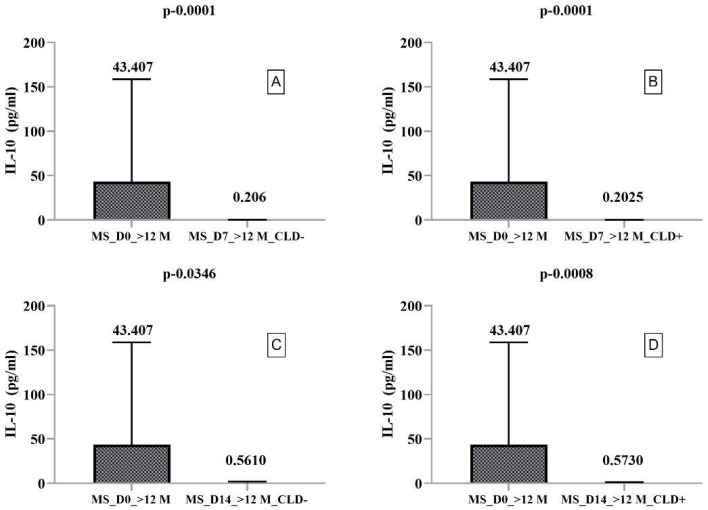
Boxplot type diagrams through which we followed the medians and interquartile range. Median values and statistical significance are mentioned in the graph. Comparison of CLD exposure effects on IL-10 secretion by PBMCs harvested from RRMS patients with a disease duration of >12 months on day 0 with days 7 and 14. (**A**) Comparison between the level of IL-10 secreted by PBMC ex vivo and on day 7 of the study in the absence of exposure to CLD; (**B**) Comparison between the level of IL-10 secreted by PBMC ex vivo and 7 days after exposure to CLD; (**C**) Comparison between the level of IL-10 secreted by PBMC ex vivo and on day 14 of the study in the absence of exposure to CLD; (**D**) Comparison between the level of IL-10 secreted by PBMC ex vivo and 14 days after exposure to CLD.

**Figure 4 ijms-23-10262-f004:**
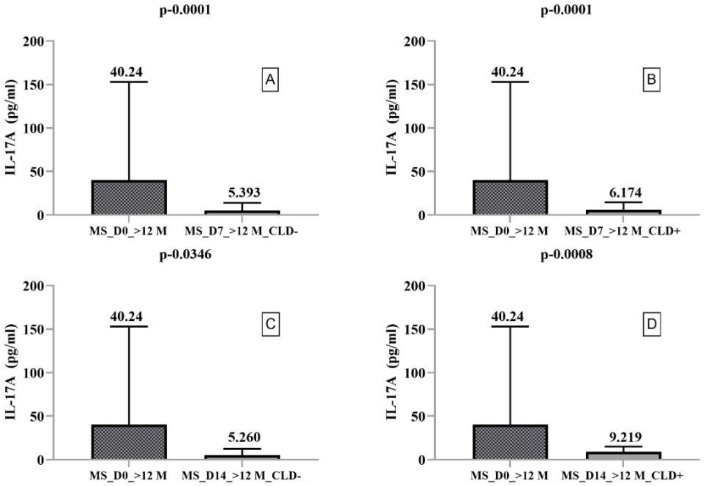
Boxplot type diagrams through which we followed the medians and interquartile range. Median values and statistical significance are mentioned in the graph. Comparison of CLD exposure effects on IL-17A secretion by PBMCs harvested from RRMS patients with a disease duration of >12 months on day 0 with days 7 and 14. (**A**) Comparison between the level of IL-17A secreted by PBMC ex vivo and on day 7 of the study in the absence of exposure to CLD; (**B**) Comparison between the level of IL-17B secreted by PBMC ex vivo and 7 days after exposure to CLD; (**C**) Comparison between the level of IL-17A secreted by PBMC ex vivo and on day 14 of the study in the absence of exposure to CLD; (**D**) Comparison between the level of IL-17A secreted by PBMC ex vivo and 14 days after exposure to CLD.

**Figure 5 ijms-23-10262-f005:**
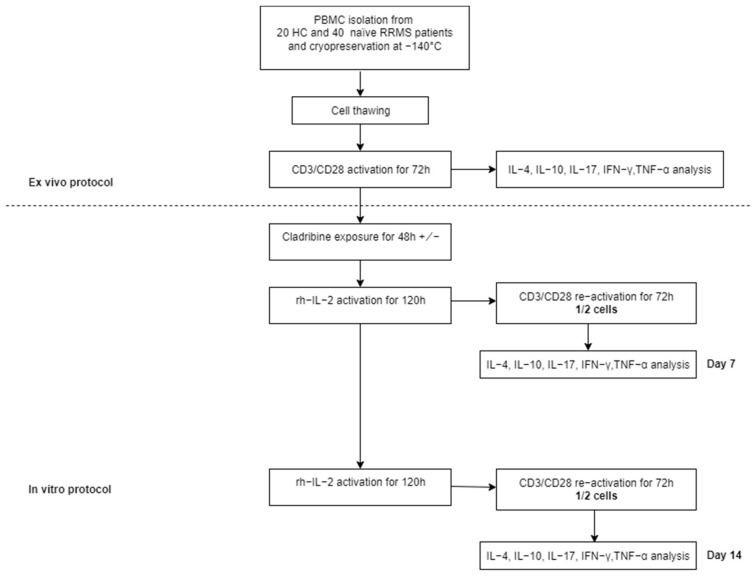
Study protocol.

**Table 1 ijms-23-10262-t001:** Demographic and disease-associated data for naïve RRMS patients and HCs.

	RRMS (*n* = 40)	RRMS ≤ 12 Months (*n* = 19)	RRMS > 12 Months (*n* = 21)	HC (*n* = 20)
Age	36.08 ± 9.59	33.85 ± 9.18	38.30 ± 9.25	35.82 ± 8.71
Age at onset	34.05 ± 8.85	32.55 ± 9.07	35.55 ± 8.59	
Sex (F/M)	27/13	14/5	13/8	13/7
Disease duration (months)	21.8 ± 23.5	5.3 ± 3.0	38.3 ± 23.5	
Number of relapses in the last year	1.38 ± 1.00	1.30 ± 0.73	1.45 ± 1.20	
Total number of relapses after disease onset	2.75 ± 0.50	1.40 ± 0.75	3.05 ± 2.06	
EDSS	2.5 ± 0.7	1.3 ± 1.0	2.5 ± 1.3	

**Table 2 ijms-23-10262-t002:** Descriptive cytokine statistics and the inflammatory ratio for HCs and RRMS patients grouped by disease duration (≤12 and >12 months) at PBMC collection (D0) and on days 7 (D7) and 14 (D14) with (+) and without (−) CLD exposure.

Sample		IL-4	IL-10	IL-17A	IFN-γ	TNF-α	Inflamatory Ratio
		pg/mL	pg/mL	pg/mL	pg/mL	pg/mL	
HC_D0	Median	6.784	48.390	109.854	1304.469	156.642	21.011
	Minimum	1.114	0.401	5.773	26.637	5.136	2.532
	Maximum	15.881	795.472	519.820	3671.328	870.336	166.842
HC_D7_CLD−	Median	1.185	0.205	2.890	15.337	1.970	17.231
	Minimum	0.477	0.016	0.763	1.732	0.049	3.644
	Maximum	455.557	631.851	199.197	975.272	2787.694	62.140
HC_D7_CLD+	Median	2.076	0.190	5.757	52.742	3.103	14.651
	Minimum	0.614	0.016	0.234	2.297	0.049	2.680
	Maximum	1150.319	681.029	557.426	1187.327	3162.460	63.784
HC_D14_CLD−	Median	7.822	0.284	4.643	64.914	10.721	9.448
	Minimum	0.587	0.016	0.155	2.547	0.095	1.881
	Maximum	126.552	9.646	48.838	799.315	174.443	80.124
HC_D14_CLD+	Median	15.977	0.246	4.131	91.888	12.897	8.353
	Minimum	0.306	0.016	0.329	1.915	0.049	3.115
	Maximum	166.812	7.330	178.369	634.519	446.454	75.199
MS_D0_≤12M	Median	4.255	94.056	64.862	1033.515	310.297	15.340
	Minimum	0.748	0.096	2.306	0.850	18.500	3.521
	Maximum	12.843	407.926	554.707	5250.620	1473.799	97.354
MS_D0_>12M	Median	7.135	43.407	40.240	580.137	103.813	16.727
	Minimum	0.748	0.124	0.793	4.139	0.049	1.226
	Maximum	23.987	291.273	346.410	4490.099	1078.030	115.659
MS_D7_≤12M_CLD−	Median	1.432	0.232	4.189	16.059	1.0367	20.920
	Minimum	0.123	0.011	0.598	4.945	0.049	5.326
	Maximum	31.854	93.441	177.527	1186.227	2454.743	114.248
MS_D7_>12M_CLD−	Median	1.513	0.206	5.393	14.616	2.270	20.926
	Minimum	0.123	0.025	0.162	1.230	0.049	3.673
	Maximum	639.767	265.128	389.493	1046.216	3283.021	45.340
MS_D7_≤12M_CLD+	Median	1.680	0.193	5.134	35.0514	1.695	20.166
	Minimum	0.123	0.007	0.427	3.054	0.049	1.488
	Maximum	48.682	22.019	107.006	1022.714	898.379	290.501
MS_D7_>12M_CLD+	Median	1.102	0.202	6.173	13.002	0.947	18.850
	Minimum	0.123	0.007	0.069	1.704	0.049	2.218
	Maximum	1245.248	286.989	418.538	1399.948	1580.564	78.493
MS_D14_≤12M_CLD−	Median	7.161	0.460	6.245	79.069	10.721	11.645
	Minimum	2.089	0.016	0.234	4.828	0.303	0.905
	Maximum	153.917	2.841	30.329	1046.216	259.611	56.422
MS_D14_>12M_CLD−	Median	14.014	0.561	5.260	114.436	19.124	6.215
	Minimum	1.162	0.132	0.314	15.505	2.631	0.865
	Maximum	454.291	61.854	86.417	454.274	87.586	107.052
MS_D14_≤12M_CLD+	Median	8.673	0.462	9.004	93.323	13.166	8.154
	Minimum	0.801	0.058	0.046	8.278	0.095	0.751
	Maximum	210.267	1.470	34.125	852.127	116.855	56.842
MS_D14_>12M_CLD+	Median	15.396	0.57278	9.21902	76.225	17.704	5.434
	Minimum	0.801	0.073	0.046	0.439	0.049	0.611
	Maximum	86.561	2.867	173.672	824.659	203.954	74.150

**Table 3 ijms-23-10262-t003:** Comparison of CLD exposure effects on IL-4 secretion by PBMCs from RRMS patients and HCs on days 7 and 14.

Dunn’s Multiple Comparison Tests	Mean Rank Diff.	Significant?	Summary	Adjusted *p*
HC_D7_CLD− vs. HC_D14_CLD−	−94.00	Yes	*	0.0275
HC_D7_CLD+ vs. HC_D14_CLD+	−73.53	No	ns	0.3308
MS_D7_CLD− vs. MS_D14_CLD−	−98.26	Yes	****	<0.0001
MS_D7_CLD+ vs. MS_D14_CLD+	−100.10	Yes	****	<0.0001

* statistically significant if *p* < 0.05; **** very highly significant if *p* < 0.0001.

**Table 4 ijms-23-10262-t004:** Comparison of CLD exposure effects on TNF-α secretion by PBMCs from RRMS patients and HCs on days 7 and 14.

Dunn’s Multiple Comparison Tests	Mean Rank diff.	Significant?	Summary	Adjusted *p*
HC_D7_CLD− vs. HC_D14_CLD−	−73.73	No	ns	0.3233
HC_D7_CLD+ vs. HC_D14_CLD+	−41.93	No	ns	>0.9999
MS_D7_CLD− vs. MS_D14_CLD−	−73.06	Yes	**	0.0074
MS_D7_CLD+ vs. MS_D14_CLD+	−63.94	Yes	*	0.0440

* statistically significant if *p* < 0.05; ** highly significant if *p* < 0.01.

**Table 5 ijms-23-10262-t005:** Comparison of CLD exposure effects on TNF-α secretion by PBMCs from RRMS patients with a disease duration of >12 months on day 0 with days 7 and 14.

Dunn’s Multiple Comparison Tests	Mean Rank diff.	Significant?	Summary	Adjusted *p*
MS_D0 vs. MS_D7_CLD−	44.23	Yes	****	<0.0001
MS_D0 vs. MS_D7_CLD+	46.35	Yes	****	<0.0001
MS_D0 vs. MS_D14_CLD−	22.45	No	ns	0.1434
MS_D0 vs. MS_D14_CLD+	26.98	Yes	*	0.0326

* statistically significant if *p* < 0.05; **** very highly significant if *p* < 0.0001.

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
