# Peer review of "Cytokine Secretion Dynamics of Isolated PBMC after Cladribine Exposure in RRMS Patients"

_ijms, 2022, doi:10.3390/ijms231810262_

Round 1

Reviewer 1 Report

Review of the manuscript entitled: Cytokine secretion dynamics of isolated PBMC after CLD exposure in RRMS patients. The manuscript is interesting but should be improved. References should be added, for example, in lines 40, 45, 59, 64, 76, 81, 283, 303,

At the end of the introduction aim of the manuscript should be added “The aim of the present study was to…” In my opinion the methodology is well described. Since the discussion is very long, I strongly recommend adding a few-sentence conclusion.

Author Response

Review of the manuscript entitled: Cytokine secretion dynamics of isolated PBMC after CLD exposure in RRMS patients. The manuscript is interesting but should be improved. References should be added, for example, in lines 40, 45, 59, 64, 76, 81, 283, 303.

Indeed, the bibliographic titles related to the text mentioned by the reviewer were omitted by mistake. We thank the reviewer for the valuable suggestions. The text has been improved.

At the end of the introduction aim of the manuscript should be added “The aim of the present study was to…” In my opinion the methodology is well described. Since the discussion is very long, I strongly recommend adding a few-sentence conclusion.

Thank the reviewer for the valuable comment. The text has been improved.

Reviewer 2 Report

In the paper, the authors have investigated the immunomodulatory role of cladribine (CLD) by evaluating cytokine production from peripheral blood mononuclear cells (PBMCs) harvested from 40 patients with untreated relapsing-remitting MS (RRMS) exposed to CLD. They found that PBMCs isolated from patients with RRMS for ≤12 months had significantly higher IL-4 but significantly lower IFN-γ and TNF-α secretion after CLD exposure. PBMCs isolated from patients with RRMS for >12 months had altered inflammatory ratios toward an anti-inflammatory profile and increased IL-4 but decreased TNF-α secretion after CLD exposure. CLD induced nonsignificant changes in IL-17 secretion in both RRMS groups.

 Major points

-The protocol is not clear enough. It seems like that they have first stimulated the PBMCs with anti-CD3/CD28 for 72h, then 48h with CLD, then 5 or 10 days with IL2, then 7 or 14 days again with anti CD3/CD28 following 20h without IL2 (lines 129-135). Therefore, it seems like that the PBMCs were in colture for up to 30 days. Is it correct? It is a very long period, and we expect a high mortality. What is the vitality of the cells throughout the experimental period?

-Why not just stimulating the cells with antiCD3/CD28 with or without CLD for a short period of time, such as 2-3 days??

-Also, did the author check the toxicity of CLD in this setting? How have they chosen the concentration of CLD? (besides, in line 131, the text reports “10–7 M CLD”- what does it mean?? Is it 10.7 Molar of CLD??)

-A statistical comparison between HC and MS patients needs to be provided for all the analyses

-What is the final aim of the study? If the authors are looking for a biomarker of CLD responsiveness, they should perform other appropriate statistical analysis (for instance ROC analysis) to evaluate sensitivity/specificity of the chosen parameters.

Minor points

-Please specify what is “NA/LE” CD3 monoclonal antibody”

-Please correct the paragraph “The Mann-Whitney U and Kruskal Wallis tests were used for intergroup comparisons of nonparametric variables, which are reported as median and range (minimum–maximum), which can be used with >2 groups.”. Only the Kruskal Wallis compares more than 2 groups

-Results can be better provided also as graphs. It would be easier to read.

-Delete paragraph 5 and 6 (lines 483-488)

Author Response

Major points

The protocol is not clear enough. It seems like that they have first stimulated the PBMCs with anti-CD3/CD28 for 72h, then 48h with CLD, then 5 or 10 days with IL2, then 7 or 14 days again with anti CD3/CD28 following 20h without IL2 (lines 129-135). Therefore, it seems like that the PBMCs were in colture for up to 30 days. Is it correct? It is a very long period, and we expect a high mortality. What is the vitality of the cells throughout the experimental period? Why not just stimulating the cells with antiCD3/CD28 with or without CLD for a short period of time, such as 2-3 days??Also, did the author check the toxicity of CLD in this setting? How have they chosen the concentration of CLD? (besides, in line 131, the text reports “10–7 M CLD”- what does it mean?? Is it 10.7 Molar of CLD??)-

Indeed, the protocol was not properly explained in the initial form of the manuscript which is why Figure 1 was replaced and changes were made in the text, for a better understanding of the study protocol. The duration of the experiment after TCR activation and exposure to cladribine was 7 and 14 days.

We thank reviewer for careful analysis of the manuscript and valuable suggestions.

Cladribine interferes with DNA synthesis and repair through incorporation into DNA and through inhibition of enzymes involved in DNA metabolism and for these reasons already TCR-activated and proliferating lymphocytes were exposed to Cladribine.

TCR-activated lymphocytes were cultured for 48h with 10 -7 M (100nM) or without cladribine and transferred to a cladribine-free medium with rhIL2 for up to 7 or 14 days. The protocol for Cladribine exposure was established according to the protocol published by Korsen et al., 2015 and following the optimization of Cladribine concentrations for an optimal ratio of cell viability/immunomodulatory effect. The aim of the protocol was to assess the specific cytokine secretion from the surviving T lymphocytes after Cladribine exposure, compared to non-exposed T lymphocytes.

A viability assay was performed using the FACSAria III flow cytometer and Becton Dickinson Horizon™ Fixable Viability Stain 780 (FVS 780, BD cat. no. 565388). Lymphocytes were analyzed for viability at day 0 (before culture initiation) and at day 7 or 14. The cytotoxic effect of cladribine was assessed by absolute CBC using a Sysmex XS 800i Hematology Analyzer. The changes in absolute lymphocyte number reflected cell proliferation or depletion in response to the cytotoxic action of cladribine. Survival indexes were established as a ratio between absolute viable cell number without or after cladribine exposure, and initial absolute viable cell number. Proliferation indexes were calculated as a ratio of absolute viable cell number before and after 7 days and 14 days of culture with rh-IL2.

The results concerning the cytotoxic effect of Cladribine on T cells were previously published by our team (Dobreanu et al., 2021). Here are briefly our findings: the  initial T cell proliferation index after 48h of cladribine exposure was lower for treated cells from both HC and RRMS patients, compared to untreated cells. The median of proliferation index was higher in RRMS patients compared to HC. These differences suggest that T cells of RRMS patients tend to proliferate (as opposed to T cells of HC) and are also more resistant to cladribine compared to T cells of HC. In both HC and RRMS, only treated T cells proliferated continuously until day 14, suggesting that T cells resistant to cladribine display a consistent proliferative behavior.

-A statistical comparison between HC and MS patients needs to be provided for all the analyses- In Results- section “Secretory profile of PBMCs harvested from all RRMS patients compared to HCs”, we presented only the results that are future clinical significant for personalized treatment. In this section we presented the effect of exposure/lack of exposure to CLD on the secretory profile of PBMC in the case of patients with RRMS and HC, without intending to compare cytokine expression between the 2 groups at different time points.

-What is the final aim of the study?

The aim of this study was to evaluate the change in the cytokine secretion profile of PBMCs harvested from naïve RRMS patients in the presence/absence of CLD, respectively, to evaluate CLD’s immunomodulatory effects on surviving PBMCs. By mistake, the aim of the study was omitted from the text. We thank the reviewer for the valuable comments.

If the authors are looking for a biomarker of CLD responsiveness, they should perform other appropriate statistical analysis (for instance ROC analysis) to evaluate sensitivity/specificity of the chosen parameters.

We did not follow the performance of a biomarker in this study, as a result we did not have to use the performance parameters (sensitivity, specificity), respectively to analyze the ROC curves, but in the future we will take this aspect into consideration. Thank you for the valuable suggestion.

Minor points

-Please specify what is “NA/LE” CD3 monoclonal antibody”

 Purified NA/LE Mouse Anti-Human CD3 (BD cat.no. 555336) means No Azide/Low Endotoxin HIT3a monoclonal CD3 antibody. It specifically binds to the human CD3ε-chain of the CD3/T cell antigen receptor complex, being mitogenic for T lymphocytes. The text has been improved.

-Please correct the paragraph “The Mann-Whitney U and Kruskal Wallis tests were used for intergroup comparisons of nonparametric variables, which are reported as median and range (minimum–maximum), which can be used with >2 groups.”. Only the Kruskal Wallis compares more than 2 groups-

The text has been corrected. Thank you for the careful analysis of the text and valuable comments.

-Results can be better provided also as graphs. It would be easier to read.

Table 5, 6, 7 and 8 were replaced with figures.

-Delete paragraph 5 and 6 (lines 483-488)-

Due to a technical error, the conclusions were included in the discussion section. We completed paragraph 5 and deleted paragraph 6. Thank you for the valuable suggestions.

Round 2

Reviewer 2 Report

The previous issues have been addressed

Author Response

Thank you for your suggestions